# Effect of Non-Pharmacological Methods in the Reduction of Neonatal Pain: Systematic Review and Meta-Analysis

**DOI:** 10.3390/ijerph20043226

**Published:** 2023-02-12

**Authors:** Inmaculada García-Valdivieso, Benito Yáñez-Araque, Eva Moncunill-Martínez, M. Jesús Bocos-Reglero, Sagrario Gómez-Cantarino

**Affiliations:** 1General Hospital Mancha Centre (HGMC), Castilla-La Mancha Health Service (SESCAM), 13600 Alcázar de San Juan, Spain; 2Department of Physical Activity and Sports Sciences, University of Castilla-La Mancha, Toledo Campus, 45071 Toledo, Spain; 3Toledo University Hospital (HUT), Neonatal and Pediatric Oncology, Castilla-La Mancha Health Service (SESCAM), Theoretical Collaborator University of Castilla-La Mancha, Toledo Campus, 45071 Toledo, Spain; 4National Hospital of Paraplegics (HNP), Castilla-La Mancha Health Service (SESCAM), Theoretical Collaborator University of Castilla-La Mancha, Toledo Campus, 45071 Toledo, Spain; 5Faculty of Physiotherapy and Nursing, University of Castilla-La Mancha, Toledo Campus, 45071 Toledo, Spain

**Keywords:** breastfeeding, kangaroo-mother care method, sucrose, suction, pain management, pain measurement, analgesia, nursing care, newborn infant

## Abstract

In neonatology, neonates have traditionally been considered incapable of feeling pain, due to the immaturity of their nervous system. Currently, there is sufficient information on the perception of pain in neonates; however, this treatment at this crucial stage for development requires a better approach. For this reason, the aim of this study was to analyse the efficacy of non-pharmacological analgesia interventions during heel prick, and to assess their effects on heart rate (HR), premature infant pain profile (PIPP) and O_2_ saturation. A systematic review and meta-analysis was performed following the guidelines of the preferred reporting items for systematic reviews and meta-analyses (PRISMA), and the Cochrane collaboration handbook. The databases PubMed, Cochrane Library, Web of Science, Scopus, CINAHL and Science Direct were searched until the end of January 2022. The DerSimonian and Laird methods were used to estimate the effect size with a 95% confidence interval (CI95%). Effect size estimates were 0.05 (95% CI: −0.19, 0.29) for HR, −0.02 (95% CI: −0.24, 0.21) for PIPP scale, and −0.12 (95% CI: −0.29, 0.05) for O_2_ saturation. The non-pharmacological interventions analysed (breastfeeding, kangaroo-mother care method, oral sucrose and non-nutritive sucking) were not statistically significant in reducing neonatal pain, but did influence the decrease in pain score and a faster stabilisation of vital signs.

## 1. Introduction

Acute pain is defined as “an unpleasant somatic or visceral sensation caused by actual or potential tissue injury” [1]. Previously, it was believed that newborns did not have the capacity to experience pain because the myelination process of the nervous system was not complete yet. There is now enough evidence for the experience of neonatal pain, as the neurophysiological and anatomical components necessary for the transmission of the pain stimulus develop before 24 weeks’ gestational age [2]. In contrast, the nociceptive inhibitory mechanisms response for reducing pain require several months to mature [3]. After the first moments of life, newborns are subjected to a series of invasive procedures, the most frequent being heel pricks for screening for endocrinometabolic diseases [4,5]. The average number of these painful interventions increases when birth occurs before 37 weeks, and these newborns considered ”preterm” require admission in neonatal intensive care units (NICU), experiencing as many as 10–15 painful procedures per day [6]. According to the World Health Organization (WHO), there are around 15 million preterm births each year, and nationally in 2021, figures from the National Institute of Statistics (INE) showed that 20,189 births were preterm. This condition is one of the main causes of perinatal mortality and morbidity [7,8].

Studies show that unrelieved acute pain can affect the neurodevelopment of neonates, producing motor, cognitive, and behavioural alterations [9,10]. It is therefore necessary to modify clinical practice and attenuate stimuli in order to guarantee the comfort of these patients [11,12]. This paradigm shift has promoted the application of new models of care such as the newborn individualized developmental care and assessment program (NIDCAP) method. One of the purposes of this method is to assess the ability of newborns to cope with stress before, during and after procedures [13,14,15]. In addition, this method establishes a series of strategies including the use of non-pharmacological measures to prevent neonatal pain. Such measures include breastfeeding, the mother-kangaroo method, oral sucrose of glucose and non-nutritive sucking, which have emerged to prevent the side effects that drugs can induce in neonates due to pharmacokinetic and pharmacodynamics differences in respect to the adult patient [16,17,18]. It is known that hepatic metabolism is physiologically decreased, because the cytochrome P450-dependent enzymes are 30–50% lower than that in adults. The glomerular filtration rate is also decreased, and until the neonate reaches 6–12 months of age, its values do not match those of the adult. This results in a slower excretion of drugs, exerting a more intense and prolonged effect. Neonates exposed to the most commonly used opioids in NICU such as Morphine, Fentanyl or Midazolam may experience respiratory depression, seizures, nausea and vomiting, urinary retention and decreased intestinal motility [12,17].

The inability of newborns to verbalize pain perception has led to the development of scales for pain assessment and evaluation. These tools are based on the observation of physiological and behavioural changes that neonates experience as a result of pain [19]. Among the most widely used validated scales is the premature infant pain profile (PIPP) scale, used to assess acute and procedural pain in preterm infants up to 28 weeks of gestation. It consists of 7 items scored from 0 to 3, with a score of 12 or more indicating severe pain [20]. Similarly, the neonatal infant pain scale (NIPS) is widely used to assess acute pain in preterm infants under 37 weeks and consists of 6 items, which are scored from 0 to 1, with a score of 4 or more indicating severe pain [21].

According to the American Academy of Pediatrics (AAP), pain reduction in newborns remains an area that needs to be better addressed and, although there are studies that have evaluated the usefulness of using non-pharmacological analgesia methods, a consensus on their application has not been reached yet [17]. Therefore, the aim of this study was to analyse the efficacy of non-pharmacological analgesia methods during procedures such as heel prick and to assess their effects on the reduction of neonatal pain through variables such as heart rate (HR), PIPP score and O_2_ saturation. The outcome measures were chosen based on their physiological effect caused by pain, due to the fact that newborns tend to suffer oxygen desaturation as well as an increased heart rate. 

## 2. Materials and Methods

This study was carried out with the preferred reporting items for systematic reviews and meta-analyses (PRISMA) guidelines [22] and following the recommendations of the Cochrane collaboration handbook [23]. The protocol has been registered in PROSPERO (registration number: CRD42022380996). 

A systematic search of PubMed, Cochrane Library, Web of Science, Scopus, CINHAL and Science Direct was conducted until the end of January 2022. After examining the types of non-pharmacological treatment, the search strategy included the following terms: (“sucrose” OR “breastfeeding” OR “suction” OR “kangaroo-mother care method”) AND (“pain management” OR “pain measurement” OR “analgesia” OR “nursing care”) AND (“newborn infant”) AND (“neonatal intensive care units”). References cited in the selected articles were also reviewed for inclusion in this review.

Studies were selected according to the following inclusion criteria, including: (a) infants whose minimum gestational age was 28 weeks, without restriction of sex or ethnic group; (b) infants whose weight was greater than 1 kg; (c) infants whose Apgar test score was greater than 5 in the first 5 min; (d) a comparison between different methods of non-pharmacological analgesia; (e) a comparison of non-pharmacological analgesia methods with no intervention; (f) an assessment of neonatal pain using the PIPP and NIPS scales; and (g) a randomized controlled trial (RCT) or quasi-experimental design.

On the other hand, the exclusion criteria were studies: (a) involving infants with congenital anomalies; (b) not written in English or Spanish; or (c) ineligible by design as narrative reviews, observational studies or case reports. The following data were extracted from the selected studies: (a) information on authors and year of publication; (b) country of intervention; (c) sample characteristics (sample size, gestational age, birth weight, Apgar test and pain measurement); and (d) intervention characteristics (type and dosage used). Interventions were classified as: breastfeeding, mother-kangaroo method, oral sucrose or glucose and non-nutritive sucking, according to their characteristics.

The risk of bias of RCTs was assessed using the Cochrane RoB2 tool [24]. This tool includes the assessment of five domains: randomization process, deviations from intended interventions, missing data on outcomes, measurement of outcomes and selection of reported outcomes. The overall bias was considered as “low risk” if the study in question scored “low risk” in all domains, “some concern” if any domain was rated as “some concern”, and “high risk” if at least one domain was rated as “high risk” or several domains were rated as “some concern”. Furthermore, the risk of bias of non-randomized clinical trials was assessed using the ROBINS-I tool [25], which includes seven domains: confounding, selection of participants, classification of interventions, deviations from intended interventions, missing data on outcomes, measurement of outcomes and selection of reported outcomes. The risk of bias was classified as “low risk” if the study in question scored low in all domains, “moderate risk” if the rating was moderate for all domains, “serious risk” if the assessment was considered serious in at least one domain, and “critical risk” if the assessment was considered critical in at least one domain. In the selection process and data collection process, three authors (I.G.-V., B.Y.-A. and S.G.-C.) independently reviewed each publication to verify that it met the stated inclusion criteria, and then decisions on which data to include in the review were made jointly. In the risk of bias assessment, the members mentioned above worked together for both randomized controlled trial and quasi-experimental studies. 

The standardized mean difference score using Cohen’s d-index was calculated for each of the neonatal pain variables (HR, PIPP scale and O_2_ saturation). The DerSimonian and Laird method was used to calculate the pooled effect size estimate with a 95% confidence interval (95% CI). In the statistical analysis, positive values indicated an increase in PIPP scores in the control group compared to the intervention group. Cohen’s d-statistic values below 0.2 indicated a weak effect, values around 0.5 indicated a moderate effect, and values above 0.8 indicated a strong effect [26]. 

The inconsistency of the results was assessed using the I^2^ statistic, and the values obtained were considered as: without important (0–30%), moderate (30–50%), substantial (50–75%) or considerable (75–100%). The p-statistic values were also evaluated. Sensitivity analyses were performed for each of the neonatal pain endpoints (HR, PIPP scale and O_2_ saturation), eliminating studies one by one in order to assess the robustness of the estimates obtained [22]. 

Publication bias was calculated by applying Egger’s asymmetric regression test [27], and a *p* < 0.10 was estimated to be statistically significant. Statistical analyses were performed using Stata V.15.1 software.

## 3. Results

The search identified 24 studies (Figure 1) [28,29,30,31,32,33,34,35,36,37,38,39,40,41,42,43,44,45,46,47,48,49,50,51] that were included in this systematic review, of which 6 were included in the meta-analysis [33,40,41,42,44,46]. With a total of 2246 participants, the studies were conducted on three continents (17 in Asia, 5 in USA, and 2 in Europe) and were published between 2011 and 2021. Included participants were between 29 and 39 weeks’ gestational age, with sample sizes ranging from 28 to 243 subjects. All interventions were performed in the NICU by trained professionals with extensive experience in newborn care. The duration of the interventions ranged from 2 to 80 min (Table 1). 

For the measurement of neonatal pain, 6 studies used the NIPS scale and 18 used the PIPP scale. In addition, the control groups were without intervention in 8 studies and with routine health interventions in another 8 studies. As for the procedures performed in the intervention group for the reduction of neonatal pain, 8 tested the analgesic effect of breastfeeding, 12 tested the mother-kangaroo method, 4 tested the oral glucose at 20–25%, 9 tested oral sucrose at different concentrations ranging from 20–50% and tested 8 non-nutritive sucking.

### 3.1. Risk of Bias in the Studies 

According to the RoB2 assessment of risk of bias in RCTs, 33.3% of the studies showed a high risk of bias, 60% showed a moderate risk of bias, and 6.7% showed a low risk of bias. Specifically, 93.3% of studies had a moderate to low risk of bias in the randomization process, 53.3% had a low risk of bias in the deviation from the intended interventions and 20% had a moderate risk of bias in the selection of outcomes (Figure 2). The evaluation of each study is shown below (Figure 3). 

In non-randomized trials, the risk of bias assessment with the ROBINS-I scale revealed that 11.11% of studies had a critical risk of bias, 22.22% had a severe risk of bias, 44.44% had a moderate risk of bias and, finally, 22.22% had a low risk of bias. Specifically, 55.55% of the studies reflected a low risk of bias in the selection of participants, 44.44% reflected a moderate to severe risk of bias in the deviation from the intended interventions, and 33.33% reflected a critical risk of bias in the selection of outcomes (Figure 4).

### 3.2. Meta-Analysis 

The pooled effect size was 0.05 (95% CI: −0.19, 0.29) for HR, with moderate inconsistency (I^2^ = 46.8%, *p* = 0.069); −0.02 (95% CI: −0.24, 0. 21) for the PIPP scale, with a non-significant inconsistency (I^2^ = 16.0%, *p* = 0.311); and −0.12 (95% CI: −0.29, 0.05) for O_2_ saturation, with a non-significant inconsistency (I^2^ = 0.0%, *p* = 0.664). The forest plot for the pooled effect size is presented in Figure 5. 

### 3.3. Sensitivity Analysis

The pooled effect size estimate was unchanged for the variables HR, PIPP scale and O_2_ saturation when items were removed one by one from the analysis.

### 3.4. Publication Bias

No significant publication bias was detected for any of the pain estimation variables, as observed in Egger’s test for HR (*p* = 0.747), PIPP scale (*p* = 0.287), and O_2_ saturation (*p* = 0.106).

## 4. Discussion

This systematic review and meta-analysis examines the effects of non-pharmacological analgesia methods in reducing neonatal pain following invasive procedures. The results show that the interventions of breastfeeding, mother-kangaroo method, oral sucrose or glucose and non-nutritive sucking are not effective in reducing neonatal pain-related parameters such as HR, PIPP scale and O_2_ saturation.

Studies show that infants who were breastfed had lower scores on the PIPP scale and a more rapid stabilization of vital signs after heel lance [29,32,34]. The lactose rich content of breast milk and essential amino acids such as L-tryptophan promote the release of endogenous opioids such as beta-endorphins, which are responsible for reducing the transmission of pain to the nervous system. Furthermore, after ingestion, cholecystokinin (CCK), a neuropeptide that exerts its calming effect by generating drowsiness, is secreted [28,31,32,33]. These effects would highlight the ability of breastfeeding as one of the most common and beneficial methods to reduce pain, although such a reduction is not considered statistically significant [30,31,33]. Our data are consistent with a previous systematic review, which found that there was insufficient evidence on the analgesic effect of breastfeeding on neonatal pain responses [52]; however, it does have a greater efficacy on HR with respect to O_2_ saturation.

In relation to the impact of the mother-kangaroo method, research reports that the groups in which this intervention was applied obtained clinically lower values in the variables studied [36,37,38], but failed to reach statistical significance [35,42,43]. These results are consistent with those obtained in other studies [53,54,55], which indicate that there were no significant differences in HR, PIPP scale and O_2_ saturation between the intervention and control groups before, during and after heel prick [39,40,41]. In the same line, our findings suggest that, although the mother kangaroo method is capable of influencing the PIPP scale score, it has been found to lack a statistically significant effect. The mechanism by which the mother-kangaroo method exerts its analgesic effect is through skin-to-skin contact along with maternal smell, bonding and heartbeat. These mechanisms induce in the newborn the production of hormones such as oxytocin, which decreases the activity of the sympathetic nervous system and is associated with a state of sleep and relaxation, resulting in greater haemodynamic stability [35,42]. 

Our data also show that oral sucrose and non-nutritive sucking generate a more effective synergistic effect than each of these interventions alone. Due to the sweet taste of sucrose and the oral–tactile stimulation of non-nutritive sucking, serotonin and endorphin are released, producing an analgesic effect lasting between 5–10 min [44,46,49,51]. Specifically, of the three variables analysed, the PIPP score achieved the best results, although this systematic review was underpowered to detect significant differences. Our statements are compatible with previously published articles, as the results on the efficacy of using this technique are not conclusive [56,57,58]. Some studies argue that the combination of oral sucrose and non-nutritive suctioning is associated with a reduction in the HR and PIPP scale and an increase in O_2_ saturation [44,47,49,59,60]. In contrast, other studies show that such an intervention does not produce statistically significant differences when compared to routine care [45,48,50]. Furthermore, questions such as optimal dose and concentration or possible long-term adverse effects remain unresolved [50,57,58].

Although our results did not find significant effects, they highlight the importance of including new strategies that act prophylactically against acute pain caused by routine care in neonates. However, such strategies may contribute to limiting the use of drugs in the NICU (thus avoiding possible side effects), as they are considered a more natural alternative to the traditional pharmacological model. However, the studies reviewed do not show strong evidence in this regard. Knowledge is limited and there are contradictory responses, so the capacity of these analgesic methods needs to be studied further to build the confidence needed for their application [29,30,31]. 

Therefore, our findings demonstrate that, although there were no statistically significant differences in HR, PIPP scale and O_2_ saturation between the groups before, during and after the interventions, lower PIPP scale scores were found in the intervention groups compared to the control groups. In this regard, some authors suggest that decreased pain scores should not always be interpreted as pain relief, as further research on brain activity involved in neonatal nociception is needed [19,21]. The evaluation of techniques that categorize neonatal pain responses remains a clinical and scientific challenge [28]. Further examination of the validity of PIPP scores is therefore warranted, as they are based on the subjective observation of the assessor and should not be considered the sole parameter of assessment [20,32,50]. 

For this reason, other articles have studied the effects of non-pharmacological interventions on variables such as crying time or serum and salivary cortisol levels, finding that the duration of crying after the procedures was longer in the control groups than in the groups in which breastfeeding, mother-kangaroo method, oral sucrose or non-nutritive sucking had been administered previously [29,33,41,44,51]. Additionally, the biochemical index of pain as measured by cortisol showed that the levels of both plasma and saliva were equivalent and significantly lower in the intervention groups [33,61]. 

The inherent limitations of our study include the high risk of bias in the included articles. Larger sample sizes and more defined gestational ages would be necessary to increase the external validity of the results. Similarly, the studies include populations with different characteristics in relation to the neurodevelopmental status of newborns, which could affect physiological and behavioural functions, leading to a clinical heterogeneity that could alter the results. Similarly, the differences in the pain measurement scale used, the type of intervention or type of control group and the dose used could have influenced the homogeneity of the studies. Finally, meta-analysis was only performed in six studies, as the rest of the articles did not include sufficient data to calculate the effect size. Due to all these issues, the findings should be interpreted with caution. 

This systematic review and meta-analysis highlights the need for a multimodal approach to prevent neonatal pain and a search for alternatives to pharmacological therapy. For nurses, as the main care providers, it implies the implementation of new methodologies in daily routines, training in the observation and assessment of neonatal pain and the creation of an individualized and humanized care plan [62,63]. For research, this review implies the identification of gaps in existing knowledge and the need for further studies to test the safety and efficacy of the proposed interventions. Future research is recommended to examine what long-term consequences untreated pain may have, or whether the impact of repeated doses of oral sucrose in the neonatal stage may predispose newborns to the development of diabetes in childhood. 

## 5. Conclusions

This study allows us to conclude that the non-pharmacological techniques analysed (such as breastfeeding, the mother-kangaroo method, oral sucrose or glucose and non-nutritive sucking) are not effective in reducing neonatal pain derived from invasive procedures such as heel prick. Despite these results, we must acknowledge their influence in decreasing pain as measured by the PIPP scale, and in the more rapid stabilization of HR and O_2_ saturation. Of all analgesic methods, breastfeeding is considered the treatment of first choice because of its safety, ease of administration and availability, in addition to its multiple benefits, which have been widely studied both nutritionally and immunologically.

## Figures and Tables

**Figure 1 ijerph-20-03226-f001:**
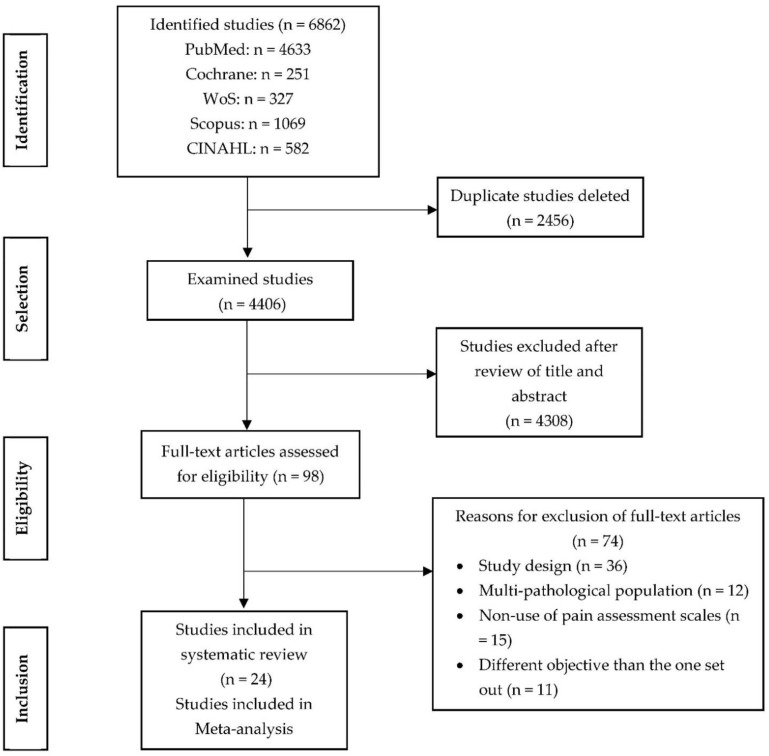
Flow chart for study selection.

**Figure 2 ijerph-20-03226-f002:**
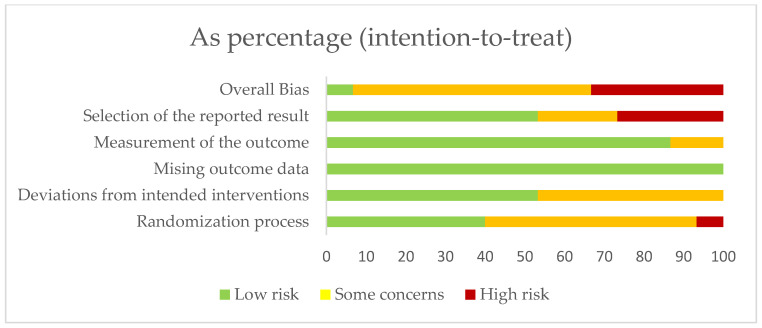
Risk of bias assessment in randomized controlled trials (RoB2).

**Figure 3 ijerph-20-03226-f003:**
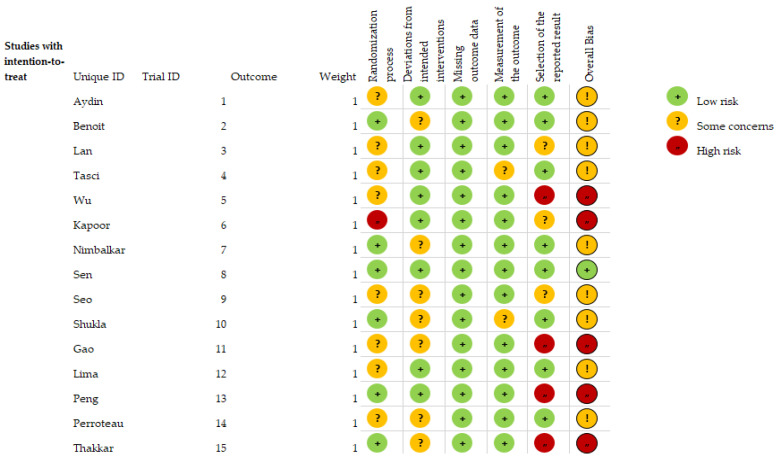
Risk of bias assessment for each included study.

**Figure 4 ijerph-20-03226-f004:**
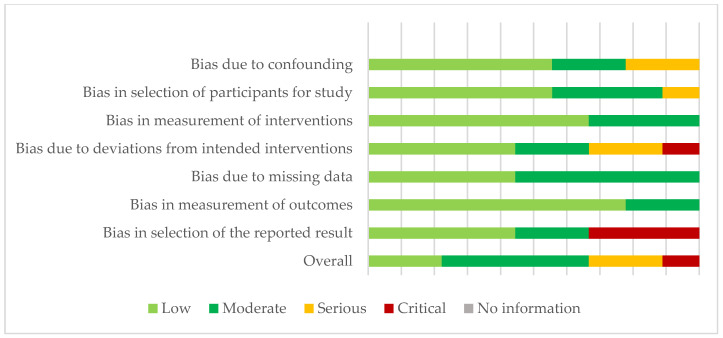
Risk of bias assessment in quasi-experimental studies (ROBINS-I).

**Figure 5 ijerph-20-03226-f005:**
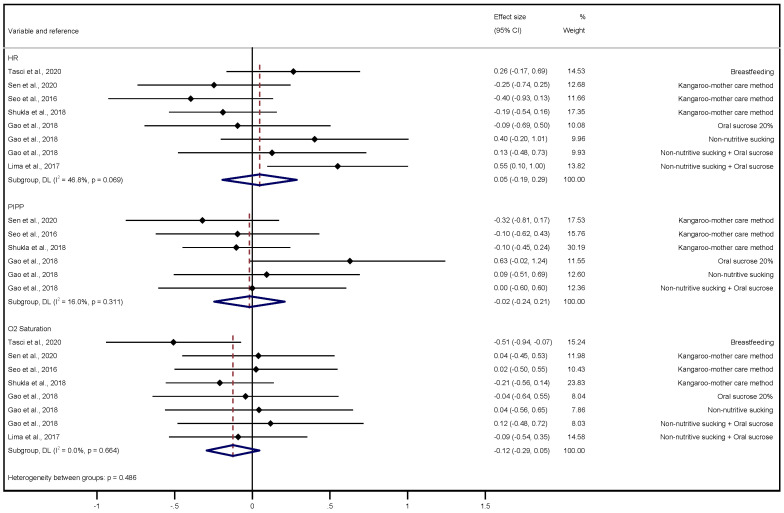
Forest plot for the pooled effect size of interventions for neonatal pain variables (HR, PIPP scale and O_2_ saturation) [33,40,41,42,44,46].

**Table 1 ijerph-20-03226-t001:** Characteristics of included studies.

Author and Year	Country	Sample Characteristics					Intervention Characteristics			
		n (Girls)	Gestational Age (Weeks)	Birth Weight (kg)	Apgar Test 5′	Pain Measurement	Intervention	Characteristics	Dose	Certainty of the Evidence GRADE
Avcin et al., 2021 [28]	Turkey	IG1: 35 (15) IG2: 35 (18) IG3: 35 (15) CG: 35 (18)	IG1: 38.48 IG2: 38.62 IG3: 38.42 CG: 38.28	IG1: 3.184 IG2: 3.293 IG3: 3.261 CG: 3.361	NA	NIPS scale IG1: 2.64/6.02/2.88 IG2: 2.85/6.20/3.03 IG3: 2.60/5.97/2.40 CG: 2.66/6.31/3.45	IG1: BF IG2: KMC IG3: Contention CG: Routine care	BF 5 min before, KMC 15 min before, lateral decubitus restraint 1 min before, during and after. NIPS scale 1 min before, during and after heel prick.	BF on demand	High
Aydin et al., 2019 [29]	Turkey	IG1: 50 (25) IG2: 50 (25) CG: 50 (25)	IG1: 39.14 IG2: 39.36 CG: 38.98	IG1: 3.344 IG2: 3.394 CG: 3.241	NA	NIPS scale IG1: 6.10 IG2: 4.44 CG: 6.42	IG1: Heel warm-up IG2: BF CG: Routine care	3–5 min before thermal bag water at 40° on the heel. 1 min before BF up to 2 min after. NIPS scale during heel prick.	BF on demand	Moderate
Bembich et al., 2018 [30]	Italy	IG1: 20 (11) IG2: 20 (12) IG2: 20 (11) IG4: 20 (8)	IG1: 39.6 IG2: 39.8 IG3: 39.8 IG4: 39.9	IG1: 3.354 IG2: 3.155 IG3: 3.429 IG4: 3.298	NA	NIPS scale IG1: 5.00 IG2: 5.50 IG3: 2.00 IG4: 2.50	IG1: Oral glucose 20% IG2: Expressed breast milk IG3: KMC + oral glucose 20% IG4: BF	2 min before glucose. Expressed breast milk 2 min before. KMC during. BF 2 min before and during the puncture. NIPS scale during heel prick.	2 mL glucose 20% 2 mL BF	Moderate
Benoit et al., 2021 [31]	Canada	IG: 18 (6) CG: 19 (8)	IG: 39.3 CG: 39.5	IG: 3.508 CG: 3.318	IG: 9 CG: 9	PIPP scale IG: 4.7/4.7/5.0/4.4 CG: 4.1/4.5/3.9/4.3	IG: BF + KMC CG: Oral sucrose 24% + NNS	BF and sucrose 2 min before. PIPP scale at 30, 60, 90 and 120 s after heel prick.	BF on demand 0.24 mL sucrose 24%	Moderate
Lan et al., 2021 [32]	Taiwan	IG2: 40 (18) IG3: 40 (20) CG: 40 (23)	IG1: 39.07 IG2: 38.91 CG: 39.42	IG1: 3.109 IG2: 3.070 CG: 3.121	IG1: 8.98 IG2: 8.98 CG: 9	NIPS scale IG1: 0.46 IG2: 0.42 CG: 0.61	IG1: BF odour + Routine care IG2: Odour + BF taste + Routine care CG: Routine care	3 min before, cotton wool impregnated with BF near the nose, removed 5 min later. 2 min before and during heel prick, BF by drip syringe. NIPS scale 5 min before, during and after.	3 mL BF	Moderate
Tasci et al., 2020 [33]	Turkey	IG: 42 (ND) CG: 42 (ND)	IG: 38–42 CG: 38–42	IG: 2.5–4 CG: 2.5–4	IG: 8 CG: 8	NIPS scale IG: 0.02/2.00/0.36 CG: 0.12/4.62/1.83	IG: BF odour CG: Formula milk odour	Filter paper impregnated with milk near the nose 3 min before, removed 9 min after. NIPS scale before, during and 2 min after heel prick.	2 mL BF 2 mL formula milk	Moderate
Wu et al., 2020 [34]	Taiwan	IG1: 34 (ND) IG2: 34 (ND) IG3: 36 (ND) CG: 36 (ND)	IG1: 32.25 IG2: 32.51 IG3: 32.33 CG: 32.64	IG1: 1.774 IG2: 1.795IG3: 1.767 CG: 1.790	IG1: 8.09 IG2: 7.91 IG3: 7.97 CG: 8.03	PIPP scale IG1: 0/13.0/7.0 IG2: 0/13.5/7.5 IG3: 0/10.0/5.0 CG: 0/14.5/8.0	IG1: BF odour or taste IG2: BF odour or taste + Heartbeat sound IG3: BF odour or taste + Heartbeat sound + NNS CG: Routine care	3 min before cotton wool impregnated with BF near the nose, removed 10 min later. 2 min before breast milk. Reproduction of mother’s heartbeat of each newborn 3 min before, up to 10 min after. Oral–tactile stimulation. PIPP scale 5 min before, during and after the heel prick.	2 mL BF	Low
Cong et al., 2011 [35]	USA	IG1: 18 (6) IG2: 10 (5)	IG1: 31.5 IG2: 32	IG1: 1.779 IG2: 1.577	NA	Escala PIPP IG1: 13.63/17.05 vs. 13.25/16.09 IG2: 8.60/10.60 vs. 9.75 14.33	IG1: KMC 80 vs. Incubator IG2: KMC 30 vs. Incubator	KMC 60 min before and 20 min after vs. 60 min incubator rest. KMC 10 min before and 20 min after vs. 30 min incubator rest. PIPP scale at 30, 60 s after heel prick.	KMC 80 and 30 min	High
Kapoor et al., 2021 [36]	India	IG1: 45 (22) IG2: 54 (23) CG: 50 (22)	IG1: 36 IG2: 36 CG: 35	NA	IG1: >5 IG2: >5 CG: >5	Escala PIPP IG1: 8.42 IG2: 8.76 CG: 13.08	IG1: KMC IG2: Oral sucrose 50% CG: Contention	KMC before and sucrose 2 min before. Supine restraint. PIPP scale 30 s after heel prick.	KMC 30 min 0.5 mL sucrose 50%	Low
Murmu et al., 2017 [37]	India	IG1: 17 (6) IG2: 17 (6) IG3: 17 (6)	IG1: 32 IG2: 32 IG3: 32	IG1: 1.824 IG2: 1.824 IG3: 1.824	IG1: 8 IG2: 8 IG3: 8	Escala PIPP IG1: 10.59 IG2: 11.24 IG3: 12.96	IG1: KMC + alternative Kangaroo + Contention IG2: Alternative Kangaroo + Contention + KMC IG3: Contention + KMC + alternative Kangaroo	KMC and alternative kangaroo before and during heel prick. Prone restraint. PIPP scale 30 s after heel prick.	KMC 30 min	Moderate
Nimbalkar et al., 2013 [38]	India	IG: 19 (ND) CG: 28 (ND)	IG: 34.02 CG: 34.02	IG: 1.730 CG: 1.730	NA	Escala PIPP IG: 5.38 CG: 10.23	IG: KMC CG: Contention	KMC before, during and after. Prone restraint 15 min. PIPP scale 30 s after heel prick.	KMC 15 +15 min	Moderate
Nimbalkar et al., 2020 [39]	India	IG1: 50 (22) IG2: 50 (23)	IG1: 33.60 IG2: 33.60	IG1: 1.604 IG2: 1.604	NA	Escala PIPP IG1: 3.40/6.98/3.57 IG2: 2.75/6.84/3.11	IG1: KMC + oral sucrose 24% IG2: Oral sucrose 24% + KMC	KMC before, during and after 1st heel prick, sucrose 2 min before 2nd heel prick, and vice versa. PIPP scale before, 1 min and 5 min after heel prick.	KMC 15 min 0.5–1 mL sucrose 24%	Moderate
Sen et al., 2020 [40]	Turkey	IG: 32 (18) CG: 32 (14)	IG: 34.38 CG: 34.95	IG: 2.102 CG: 2.244	NA	Escala PIPP IG: 4/7/3 CG: 5/7/4	IG: KMC CG: Oral sucrose 24%	KMC before heel prick and sucrose 2 min before. PIPP scale before, during and 2 min after heel prick.	KMC 15 min 0.5 mL sucrose 24%	High
Seo et al., 2016 [41]	South Korea	IG: 26 (10) CG: 30 (17	IG: 38.66 CG: 38.36	IG: 3.203 CG: 3.043	IG: 8.7 CG: 8.9	Escala PIPP IG: 2.0/4.1/4.5/2.9 CG: 2.4/6.3/9.8/8.2	IG: KMC CG: Routine care	KMC before, during and after. PIPP scale before, during, 1 min and 2 min after heel prick.	KMC 10 + 3 min	Moderate
Shukla et al., 2018 [42]	India	IG: 50 (18) CG: 50 (30)	IG: 31.68 CG: 33.90	IG: 1.460 CG: 1.780	NA	Escala PIPP IG: 7.74 CG: 8.1	IG: KMC CG: Oral sucrose 24%	KMC before, during and after. Sucrose 2 min before. PIPP scale 30 s after heel prick.	KMC 10 min 0.2 mL sucrose 24%	Moderate
Shukla et al., 2021 [43]	India	IG1: 32 (ND) IG2: 32 (ND)	IG1: 34.28 IG2: 34.28	IG1: 1.665 IG2: 1.665	NA	Escala PIPP IG1: 3.20/8.59/3.80 IG2: 3.02/8.27/3.94	IG1: KMC + Father kangaroo IG2: Father kangaroo + MMC	KMC before and during 1st heel prick, parent-kangaroo before and during 2nd heel prick, and vice versa. PIPP scale before, 1 and 5 min after.	KMC 15 min	Moderate
Gao et al., 2018 [44]	China	IG1: 22 (7) IG2: 21 (11) IG3: 22 (8) CG: 21 (8)	IG1: 31.9 IG2: 31.7 IG3: 32.0 CG: 31.3	IG1: 1.767 IG2: 1.780 IG3: 1.697 CG: 1.682	IG1: 8.8 IG2: 8.9 IG3: 8.8 CG: 8.7	PIPP scale IG1: 9.3/6.8 IG2: 10.1/7.4 IG3: 4.4/3.0 CG: 13.3/10.6	IG1: NNS IG2: Oral sucrose 20% IG3: NNS + Oral sucrose 20% CG: Routine care	NNS and sucrose 2 min before. PIPP scale 15 s before and 30 s after heel prick.	NNS 2 min 1 mL sucrose 20%	Low
Kumari et al., 2017 [45]	India	IG1: 47 (29) IG2: 47 (26)	IG1: 35.56 IG2: 35.53	IG1: 2.417 IG2: 2.685	IG1: >7 IG2: >7	PIPP scale IG1: 1.13/3.0/1.7/2.0 IG2: 1.49/2.74/1.72/1.74	IG1: Oral sucrose 24% IG2: Oral glucose 25%	2 min before sucrose or glucose. PIPP scale before, 30 s, 1 min and 2 min after heel prick.	1 mL sucrose 24% 1 mL glucose 25%	High
Lima et al., 2017 [46]	Brazil	IG: 40 (24) CG: 38 (18)	IG: 39.0 CG: 39.05	IG: 3.300 CG: 3.230	IG: 9.8 CG: 9.7	NIPS scale IG: 3.3 CG: 5.6	IG: Oral glucose 25% CG: NNS	Glucose 2 min before. NNS before and during. NIPS scale 2 min before and during heel prick.	NNS 2 min 2 mL glucose 25%	Moderate
Peng et al., 2018 [47]	Taiwan	IG1: 37 (ND) IG2: 36 (ND) CG: 36 (ND)	IG1: 31.28 IG2: 31.31 CG: 31.16	IG1: 1.572 IG2: 1.559 CG: 1.556	IG1: 7.38 IG2: 7.32 CG: 7.28	PIPP scale IG1: 2.6/4.3/1.4 IG2: 2.6/3.6/1.1 CG: 5.6/6.9/2.6	IG1: NNS + BF IG2: NNS + BF + Contention CG: Routine care	NNS y BF 2 min before. PIPP scale 10 min before, during and 10 min after heel prick.	NNS 2 min 0.5–2.0 mL BF	Low
Perroteau et al., 2018 [48]	France	IG: 30 (15) CG: 29 (16)	IG: 29.0 CG: 30.0	IG: 1.330 CG: 1.280	NA	PIPP scale IG: 6.0/12.0 CG: 7.0/13.0	IG: NNS + Contention CG: NNS	NNS after. Contention before, during and after. PIPP scale 15 s before and 30 s after heel prick.	NNS 3 min	Moderate
Silveira et al., 2021 [49]	Brazil	IG1: 34 (17) IG2: 34 (17) IG3: 34 (17)	IG1: 33.48 IG2: 33.48 IG3: 33.48	IG1: 1.851 IG2: 1.851 IG3: 1.851	IG1: ≥7 IG2: ≥7 IG3: ≥7	PIPP scale IG1: 3.3/5.5 IG2: 3.3/4.6 IG3: 4.0/4.2	IG1: NNS IG2: Oral glucose 25% IG3: NNS + Oral glucose 25%	NNS before and during, frequency > 32 suctions/min. Glucose 2 min before. PIPP scale 30 s before and 5 min after heel prick.	NNS 2 min 1 mL glucose 25%	High
Stevens et al., 2018 [50]	Canada	IG1: 81 (44) IG2: 81 (32) IG3: 81 (41)	IG1: 32.6 IG2: 32.5 IG3: 32.7	IG1: 2.002 IG2: 1.933 IG3: 2.055	NA	PIPP scale IG1: 6.8/7.0 IG2: 6.8/6.9 IG3: 6.7/6.7	IG1: 0,1 mL oral sucrose 24% IG2: 0.5 mL oral sucrose 24% IG3: 1 mL oral sucrose 24%	Sucrose and NNS 2 min before. PIPP scale at 30 s and 60 s after heel prick.	NNS 2 min 0.1–0.5–1 mL sucrose 24%	High
Thakkar et al., 2016 [51]	India	IG1: 45 (25) IG2: 45 (20) IG3: 45 (18) CG: 45 (14)	IG1: 38.68 IG2: 38.66 IG3: 38.8 CG: 38.8	IG1: ≥2.200 IG2: ≥2.200 IG3: ≥2.200 CG: ≥2.200	NA	PIPP scale IG1: 7 IG2: 9 IG3: 3 CG: 13	IG1: Oral sucrose 30% IG2: NNS IG3: Oral sucrose 30% + NNS CG: Routine care	Sucrose and NNS 2 min before. PIPP scale after heel prick.	NNS 2 min 2 mL sucrose 30%	Low

Abbreviations: Intervention group (IG); control group (CG); breastfeeding (BF); kangaroo-mother care (KMC); non-nutritive sucking (NNS); neonatal infant pain scale (NIPS); premature infant pain profile (PIPP); not available (NA).

## Data Availability

Not applicable.

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
