# Peer review of "Effect of Non-Pharmacological Methods in the Reduction of Neonatal Pain: Systematic Review and Meta-Analysis"

_ijerph, 2023, doi:10.3390/ijerph20043226_

Round 1
Reviewer 1 Report
Thank you for the opportunity to review this very interesting manuscript. The topic of this systematic review and meta-analysis is extremely interesting and of great importance.
I do have a few comments.
Abstract:
- The last sentence is quite confusing: ' not effective in reducing neonatal pain, but do influence pain scores...'. I advise rephrasing.
Introduction:
- The authors mention the negative effects of pain. The potential negative effects of analgesia are not mentioned. I would add a few sentences about the potential negative effects of analgesia. This strengthens the need for non-pharmacological interventions.
- Please explain why is chosen for O2 saturation as outcome measure
Materials and methods:
- Was a librarian involved in the search strategy?
- Why was non-pharmacological treatment/analgesia not included in the search strategy?
- Why above 1 kg? Please give more explanation regarding your inclusion criteria.
- Why were only specific non-pharmacological interventions included and not all available in literature (i.e. massage for example)?
- Only during heel pricks or also other invasive interventions? In the abstract heel pricks are mentioned, but I could not find in in the methods or results.
Results:
- Fig. 3; also an appendix available with the information per study?
Author Response
Consulte el archivo adjunto.
Muchísimas gracias.

Reviewer 2 Report
Comments
The authors performed a meta-analysis about the effect of non-pharmacological methods in reducing neonatal pain. Research on neonatal pain is conducted all over the world, and the importance of non-pharmacological methods is increasing. Therefore, this study is interesting. However, there are some concerns and requires considerable revision, as listed below.
1. The authors should preform meta-analysis according to the latest version of PRISMA and Cochrane handbook. Pef22 is not the latest version at least. PRISMA 2020 check list is helpful.
2. In the selection process and data collection process, it is unclear how many reviewers performed these tasks and whether they worked independently.
3. It the risk of bias assessment, it is unclear how many reviewers performed these tasks and whether they worked independently.
4. This study lacks certainty of evidence in the result part.
5. The authors should present registration information of the protocol (e.g. register name, registration number) or state the protocol was not registered.
6. The authors should show assessments of risk of bias for each included study.
7. Figure 4 lacks information about the results of each study. (e.g. mean and standard deviation of PIPP in non-pharmacological method and control group.)
Round 2
Reviewer 2 Report
Thank you for taking your time to revise the manuscript.
1. Authors should add sentences about how many reviewers performed the selection process, data collection process, and the risk of bias assessment in the method. They also should add the sentences about whether these tasks work independently. Authors should refer to PRISM 2020 checklist.
2. Authors should grade the certainty of the evidence according to Chapter 14: Completing ‘Summary of findings’ tables and grading the certainty of the evidence | Cochrane Training.
Author Response
Please see the attachment.
Thank you very much for your recommendations.
